# Prevalence of *Varroa destructor* in Honeybee *(Apis mellifera)* Farms and Varroosis Control Practices in Southern Italy

**DOI:** 10.3390/microorganisms11051228

**Published:** 2023-05-06

**Authors:** Roberto Bava, Fabio Castagna, Ernesto Palma, Carlotta Ceniti, Maurizio Millea, Carmine Lupia, Domenico Britti, Vincenzo Musella

**Affiliations:** 1Department of Health Sciences, University of Catanzaro Magna Græcia, 88100 Catanzaro, Italy; 2Interdepartmental Center Veterinary Service for Human and Animal Health, University of Catanzaro Magna Græcia, CISVetSUA, 88100 Catanzaro, Italy; 3Institute of Research for Food Safety & Health (IRC-FISH), Department of Health Sciences, University of Catanzaro Magna Græcia, 88100 Catanzaro, Italy; 4Nutramed S.c.a.r.l., Complesso Ninì Barbieri, Roccelletta di Borgia, 88021 Catanzaro, Italy; 5ARA Calabria (Calabria Regional Breeders Association), Via Umberto Boccioni, 88046 Lamezia Terme, Italy; 6National Ethnobotanical Conservatory, Castelluccio Superiore, 85040 Potenza, Italy; 7Mediterranean Ethnobotanical Conservatory, Sersale (CZ), 88054 Catanzaro, Italy

**Keywords:** *Varroa destructor*, infestation degree, varroosis, *Apis mellifera*, survey, varroosis control, good beekeeping practices (GBPs), integrated pest management (IPM)

## Abstract

The majority of honeybee farms in industrialized countries currently base their *Varroa destructor* control programs on the use of acaricides in conjunction with other management practices. However, the outcomes of these practices are often misunderstood and have only been studied to a limited extent. Better yields are guaranteed by having hives with low infection levels in the spring. Therefore, it is crucial to understand which beekeeping practices can result in increased control effectiveness. This study aimed to analyze the potential effects of environmental factors and beekeeping practices on the dynamics of *V. destructor* population. Experimental evidence was obtained by interpolating percentage infestation data from diagnoses conducted on several apiaries in the Calabria region (Southern Italy) with data acquired from a questionnaire on pest control strategies. Data on climatic temperature during the different study periods were also taken into account. The study was conducted over two years and involved 84 *Apis mellifera* farms. For each apiary, the diagnosis of infestation was made on a minimum of 10 hives. In total, 840 samples of adult honeybees were analyzed in the field to determine the level of infestation. In 2020, 54.7% of the inspected apiaries tested positive for V. destructor, and in 2021, 50% tested positive, according to a study of the field test findings (taking into account a threshold of 3% in July). A significant effect of the number of treatments on parasite prevalence was found. The results showed a significant reduction in the infestation rate in apiaries that received more than two treatments each year. Furthermore, it was shown that management practices, such as drone brood removal and frequent queen replacement, have a statistically significant impact on the infestation rate. The analysis of the questionnaires revealed some critical issues. In particular, only 50% of the interviewed beekeepers diagnosed infestation on samples of adult bees, and only 69% practiced drug rotation. In conclusion, it is only possible to maintain the infestation rate at an acceptable threshold by implementing integrated pest management (IPM) programs and using good beekeeping practices (GBPs).

## 1. Introduction

In the Calabria region (southern Italy), the agro-sylvo-pastoral and zootechnical sectors represent two strategic assets for socioeconomic development [1]. Among the particularly developed zootechnical sectors is beekeeping [2]. Beekeeping provides the crucial ecosystem service of pollination in addition to a variety of products valued for their nutraceutical benefits (honey, pollen, royal jelly, etc.) [1,2,3]. In the region of Calabria, where this research was carried out, it is clear that the pollination service is given special consideration. The Regional Rural Development Program (2014–2022—programming period), through Intervention 10.1.9, intended to support beekeeping practices aimed at protecting biodiversity [3]. To do this, it was decided to allocate a financial contribution for each hive that the beekeeper dedicates to pollination through nomadism in areas with plant species of lesser nectar value. Therefore, it was decided to reward the most important but less visible activity carried out by honeybees. Despite growing recognition of the value of this pollinator, the beekeeping sector continues to be decimated by hive mortality worldwide [4]. Losses continue to be significant even in Italy, where the beekeeping industry has grown steadily over the last decade [5]. These losses can be attributed to several factors [6,7,8,9,10,11,12], but the parasite *Varroa destructor* (Anderson and Trueman) and related vector-borne infectious diseases perform a significant role [13]. Honey bee colonies can become severely weakened or even die due to the burden of infectious diseases and associated agents [14]. These colony losses have occurred since *V. destructor* first appeared on *Apis mellifera*. Initially, the nefarious consequences of the species jump were successfully stemmed mainly through the use of synthetic acaricides. However, parasite-related mortality has now increased once more. This increase is a result of widespread drug resistance [15]. High frequency of treatment, prolonged use of the same drug class, under-dosing, and off-label use of antiparasitic drugs are the main causes of resistance phenomena. To reverse this trend, the scientific community has begun to view so-called good beekeeping practices (GBPs) as crucial. The GBPs are all those interconnected practices used by beekeepers to maximize apiary productivity to protect the consumers, honeybees, and the environment [16]. These practices must be accompanied by the application of biosecurity measures, such as the quarantine of swarms of unknown origin [16]. Moreover, with the environment in mind, the application of the above measures must be included in integrated pest programs (IPM). IPM is a decision-making process involving the coordinated adoption of different strategies to optimize pest management in a way that is both economical and environmentally responsible. [17]. It involves managing pests by keeping them below the level beyond which they begin to do damage rather than trying to exterminate as many as possible [18]. Among the elements of IPM programs, monitoring the level of infestation is particularly important [19]. Monitoring infestation levels allows for timing adjustments for treatments and helps keep infestation levels below the damage threshold. As a result, viral outbreaks related to parasitosis can be prevented and production levels optimized [20]. In addition, an accurate diagnosis makes it possible to establish common and shared lines of action over a territorial area. The epidemiology of *V. destructor* is not simple; it is complicated by several variables that are often active simultaneously [21]. For example, geographical elements, such as altitude, can influence levels of *V. destructor* infestation [22]. Additionally, beekeeping practices can affect pest population dynamics. Honeybee colonies are more likely to exceed the damage threshold in spring (2% of adult honeybees) when drone brood removal is not carried out [23,24,25].

Therefore, it is important to understand the parasite’s ecology, its interactions with the environment, and potential climatic, territorial, and other factors that influence its spread. It is also necessary to know the most used techniques to assist the beekeeper in breeding activities. In this direction, the Calabria region represents a particularly interesting area of investigation.

The region has a long and narrow geographic conformation, making it possible to go from the plain to the mountains in just a few kilometers. Given the variety of landscapes and land use in relation to different altitudes and plant biodiversity, it is of particular interest to examine the relationship between levels of *V. destructor* infestation, the environment, and beekeeping practices. Therefore, also because data on the prevalence of *V. destructor* infestation in Calabria are scarce and currently little is known, it was decided to conduct an epidemiological study along with the administration of a survey in different agro-ecological areas of the region. Current pest control strategies were evaluated, and parasite prevalence was correlated with them and altitude. This makes it possible to pinpoint risk factors for high infestation rates in the spring when honeybees must be in good health to produce their annual crop of honey. The information gained from this study will be useful to guide beekeepers in planning better and more effective *V. destructor* control plans.

## 2. Materials and Methods

### 2.1. Study Area

The study was conducted in the Calabria region of southern Italy. This region is characterized by a particular territory with severe landscape variations in a very tight area. Calabria region, which has an area of 15,080 km^2^ and a coastline of 738 km on the Ionian and Tyrrhenian seas, is located in the southern part of the Italian peninsula. It is one of the most mountainous regions in Italy: 42% of the territory is mountainous (altitude above 500 masl), 49% is hilly (altitude between 50 and 500 masl), only 9% is flat (altitude below 50 masl) [26]. Due to its location and hilly landscape, Calabria has a generally dry subtropical summer climate, commonly called the Mediterranean climate [27]. Mild winters and hot, rainfall-free summers are typical of coastal areas. In particular, the Ionian side experiences high temperatures and brief but intense rainfall, as it is affected by African currents [28]. One of the five provinces that make up the area of Calabria is Catanzaro. This province spans a total area of 5200 square kilometers (2000 sq mi). It is located in the center of Calabria and is surrounded by the Tyrrhenian Sea to the west, the Sila to the north, the Ionian Sea to the east, and the Calabrian Serre to the south.

### 2.2. Study Time

The study was carried out in 2020 and 2021 in the apiaries of Catanzaro province (Calabria region of southern Italy), located in the plains, hills, and mountains. Specifically, as in Giacobino et al. (2016) [29], four collections of adult honeybee samples were carried out in each research year for subsequent field diagnosis. The first sample was taken before the summer acaricidal treatment, and the second after the treatment. A third sample was taken before starting the winter acaricidal treatment; the fourth after the winter treatment.

The first and third samplings were performed to assess the hive’s parasite load before the winter and summer treatments. The second and fourth sampling (post-treatment) were made to verify the acaricidal efficacy.

Concerning climatic conditions, the average temperatures for the months before the sampling were recorded. These months coincide with the time of maximum mite population development [30].

### 2.3. Study Design and Sample Size

The apiaries were selected following a careful analysis of the data extracted from the national beekeeping registry. Out of a total of 100 beekeeping farms in the province, 60 were selected. Both apiaries with a high and low number of hives were selected. The smallest apiary had no fewer than 20 hives. The largest had a maximum number of 60 hives. As overcrowding of colonies could increase the chance of the parasite spreading, it was preferred not to select apiaries with more hives [31].

Forty-two honeybee farms were involved out of these 60 hives, in relation to the fact that the hives had to have received their last acaricide treatment at least 2 months before sampling and analysis. All of the apiaries were situated in what could be referred to as study districts, which included places with a variety of land uses and altitudinal levels.

The apiaries were geo-referenced. All the apiaries were located at altitudes between 100 to 900 m above sea level (masl). Therefore, each research district was characterized by a specific altitude stratum. Four strata were identified according to altitude: low (100–350 masl), medium (350–700 masl), and high (700–900 masl). In the apiaries that were located within each stratum, ten hives were sampled and processed for infestation diagnosis.

In summary, 42 apiaries (see Figure 1) located in the province of Catanzaro were evaluated to settle the parasitic load in July (before summer treatment) and October (before winter treatment). These operations were carried out for two years. Ten hives were randomly selected for each apiary. In total, 420 hives per year were examined.

The study was conducted in collaboration with the Calabria Regional Breeders Association (ARA Calabria). The honeybee farms examined were in 41 municipalities in the Province of Catanzaro.

The municipalities, in relation to altitude above sea level, were divided into three study groups: A1: 100–350 masl; A2: 350–700 masl; A3: 700–900 masl (Table 1).

Figure 1 shows the apiaries where adult honeybees were sampled in the summer and winter to assess the *V. destructor* load. The map was created using the open-, source software QGIS (version 2.18).

### 2.4. Data Collection and Parasitological Diagnosis

Of the various diagnostic methods that can be performed in the field, it was decided to sample a pool of adult honeybees that were then processed using the sugar roll test. This technique was preferred to the use of washing solutions or carbon dioxide (CO2). As emphasized by Bak et al. (2009) [32], there is no statistically significant difference between the diagnosis of infestation carried out with the sugar roll test and that carried out with washing solutions. Moreover, as pointed out by Bava et al. (2022) [33], the sugar roll test is a more ethical method that respects the health of the honeybee sample. Therefore, also considering the preference of the beekeepers involved, it was decided to use the sugar roll test to diagnose infestation.

Consequently, a sample of 300 honeybees was taken from each hive and processed for diagnosis by the sugar roll test. For this purpose, the *Varroa EasyCheck* device was utilized. To obtain a homogeneous sample, the honeybees were shaken from two brood honeycombs directly into a large container and mixed. From this source, a sample of subjects was then collected in a small container (urine collection container).

The samples collected in the urine container were weighed with an electronic balance (sensitive to 0.1 g) before each test to ensure an equal number of honeybees within the container. Subsequently, a full tablespoon (approximately 45 g) of powdered sugar was placed inside the transparent bowl of the device. From the first container (urine collection container), the 300 honeybees were transferred into the transparent bowl of the *EasyCheck* device. Afterward, the white basket of the device was placed upside-down back into the transparent bowl. Finally, the yellow lid was screwed on. The newly closed device was gently rotated for one minute to evenly coat all the honeybees with powdered sugar. After that, it was left to stand for an additional minute. After this time, the lid was removed, and the *Varroa EasyCheck* was turned upside down and shaken over a large container with a small amount of water to dissolve the sugar. The device was shaken until no more mites came out. Finally, the parasites were counted [20,34]. To calculate the percentage of infected individuals in the sample of adult honeybees, the number of mites was divided by the total number of honeybees, and the result was multiplied by 100 to obtain the infection rate per colony. Hives were considered positive when the percentage of infestation in the tested sample exceeded 3% [24]. The infestation rate was calculated for each apiary by averaging the results obtained for the ten hives investigated. Finally, the detected infestation rates were compared for the different sampling areas (A1, A2, A3).

### 2.5. Survey

At the time of field diagnosis, a questionnaire was administered to the beekeepers to inquire about practices commonly implemented. The questionnaire consisted of 30 questions and was divided into 3 sections.

The first section of the questionnaire collected information on the general characteristics of the beekeeping farm (e.g., geographic location, number of colonies, winter mortality per year, etc.).

The second section concerned the management practices commonly carried out (monitoring of infestation levels, queen replacement, nuclei formation, and drones brood removal); the third section concerned acaricide treatments against the *Varroa* mite (active ingredients used, drug administration methods, dates of treatments, and chemical rotation over the last three years).

Concerning the management practices analyzed, it was asked whether hive colors were alternated, whether drones’ brood removal was carried out routinely, whether feed was supplemented, the frequency of queen replacement, the diagnostic techniques used for *Varroa* infestation assessment (carbon dioxide, sugar shake, and washing solutions), and the frequency with which infestation diagnosis was carried out. Additionally, practices were examined in light of overt clinical indicators of the infestation, such as bees with deformed wings, smaller bees, the presence of honeybees with neurological symptoms, or unusual swarming in August, September, and October. The usage and administration of veterinary drugs were our final areas of emphasis. The number of annual treatments carried out, molecules used in the summer and winter, any combination of active ingredients, the rotation of veterinary drugs, and the eventual diagnosis of infestation before and after treatment are all factors considered. Finally, the beekeeping practices implemented were evaluated in relation to the recorded infestation rates.

### 2.6. Statistical Analysis

The Microsoft Excel spreadsheet (Microsoft Corporation, Redmond, WA, USA) was used to collect the information from the field sampling. After editing, the data were transferred to JASP software (version 0.16.3, JASP Team, University of Amsterdam, Amsterdam, The Netherlands) for analysis. Using descriptive statistics, variations in prevalence within parameters and the statistical significance of the relationship between risk variables and infection were assessed. ANOVA was used to compare three or more groups, including infestation percentage, altitude, and beekeeping practices. The results were considered significant for values of *p* = 0.05 or higher.

## 3. Results

### 3.1. Statistical Analysis (Parasitological Diagnosis)

Considering the infestation rate of 3% as relevant, out of the 42 apiaries examined, 23 (54.7%) were positive in the summer of 2020 and 21 (50%) in the following year. The winter diagnoses found 25 (59.5%) apiaries positive in 2020 and 24 (57.1%) in 2021. Concerning the altitude bands above sea level, 50% of the apiaries in area A1 (150–350 m) were positive in the summer of 2020 and 57.1 in 2021; 57.1% were positive in A2 (350–700 m) in both 2020 and 2021; 57.1% and 35.7% were positive in area A3 (700–900 m) in the summer 2020 and 2021, respectively. In winter 2020, a prevalence of 92.8% and 64.2% in 2021 was found in District A1; 57.1% (2020) and 64.2% (2021) in District A2; 28.5 (2020) and 42.8 (2021) in District A3. The comparison of infestation rates between the three districts showed no statistically significant difference (*p* > 0.05).

The results of the *V. destructor* diagnosis with the number of positive apiaries, prevalence (%), and percentage of infestation are shown in the table below (Table 2).

All infestation data from the summer of 2020 were merged and compared with the 2021 diagnoses; the same operation was carried out for the winter of 2020 and 2021. No statistically significant difference (*p* > 0.05) was found in this analysis.

When comparing the average percentage of infestation recorded before and after summer and winter treatment in both 2020 and 2021, a statistically significant difference (*p* 0.01) was observed (Table 3).

The comparison of the infestation rates recorded by the honeybee farms in the A1, A2, and A3 altitudinal bands did not show a statistically significant difference (*p* > 0.05). The average temperatures of the months preceding the sampling are shown below (Figure 2 and Figure 3). Figure 2 (data collected and exported from the ARPACAL—Calabria Environmental Protection Agency) [35] shows the Mediterranean climate temperatures recorded during the study period. In June 2020, coastal areas with a view of the Ionian and Tyrrhenian seas saw average temperatures between 21 and 24 degrees Celsius. On the other hand, temperatures between 12 and 21 degrees Celsius were recorded in inland areas.

In the regions bordered by the Tyrrhenian and Ionian seas, July 2020 temperatures ranged from 24 to 30 degrees Celsius, while inland areas saw temperatures between 15 and 24 degrees Celsius. The following year’s average June temperatures on the coastlines bordering the Ionian Sea were between 24 and 27 degrees Celsius, while those bordering the Tyrrhenian Sea ranged between 21 and 24 degrees Celsius (Figure 3).

In inland areas, the average temperature ranged from 24 to 28 degrees Celsius. The Ionian Sea’s coasts experienced temperatures between 24 and 30 degrees Celsius in July, while the Tyrrhenian Sea’s coasts experienced temperatures between 24 and 27 degrees Celsius. The temperature ranged from 21 to 24 degrees Celsius on the inside. In August 2020, temperatures averaged between 21 and 30 degrees Celsius on the Ionian coast and between 24 and 30 degrees Celsius on the Tyrrhenian coast. On the other hand, temperatures between 15 and 24 degrees Celsius were recorded in inland areas. The following year, temperatures along the Tyrrhenian coast ranged from 24 and 30 degrees Celsius, while those on the Ionian coast ranged from 24 to 33 degrees Celsius. Finally, temperatures in the inland areas ranged from 15 to 27 degrees Celsius.

### 3.2. Survey Results

As can be seen from Table 4 (general characteristics), twenty-two interviewed beekeepers have been conducting beekeeping activity for more than five years; the remaining twenty have been beekeepers for less than five years.

The apiaries visited were mostly (27 out of 42) larger than 50 hives. Only 21 (50%) of respondents said they alternated the color of their hives, a fundamental practice to reduce drift phenomena.

Regarding management practices (see Table 5), at the beginning of the honey yield season, some beekeepers (30.9%) systematically replace the queens of their colonies. Only 13 (29.5%) beekeepers routinely remove drone broods to lower the infestation rate. Moreover, what is very relevant is the data on such an essential practice as diagnosis: only 21 (50%) beekeepers routinely used to diagnose infestation on a sample of adult honeybees, either using the sugar roll test or an alcohol wash. However, only around 50% of those who diagnose infestation on adult honeybees sample an adequate number of hives within the apiary to have any particular statistical power. In fact, less than eight hives are frequently sampled [24]. Twelve beekeepers claim to be unaware of the sugar roll test or alcohol wash for diagnosis. These beekeepers admit that they do not use any method for determining the degree of infestation. In this case, the diagnostic procedures are limited to simple evidence of mites in the dispersal phase, the observations of honeybees with deformed wings, and the verification of the natural fall on the diagnostic bottom board. Therefore, the control plans are not based on an accurate diagnosis but rather on scheduled treatments. Typically, all interviewed beekeepers state that they provide a carbohydrate supply in autumn and spring (often in the form of sucrose syrup).

With regard to pharmaceutical treatments (see Table 6), 27 beekeepers (64.1%) perform 3 or more treatments per year, whereas the remaining 15 (35%) only perform 2 treatments. The latter figure is associated with a very common behavior: many beekeepers use an additional treatment in the springtime, specifically in May, in between the two main blooms in the area. This additional treatment usually consists of an oxalic acid drip carried out even if there is a brood present. Other beekeepers resort to the administration of formic acid products for additional treatment.

For summer treatment, 13 beekeepers (30.9%) use pharmaceutical preparations based on amitraz as the active ingredient; 15 (35.7%) perform treatments with oxalic acid dripped after caging the queen; 5 beekeepers (11.9%) employ formic acid products; 2 beekeepers (4.7%) use monoterpene-based products, such as thymol; 1 (2.3%) uses repeated oxalic acid sublimation; 6 (14.2%) use amitraz-based products, applying them after dripping oxalic acid without having previously caged the queen. At the end of the honey yield season, professional beekeepers often report using the brood removal technique, which allows them to create nuclei and simultaneously carry out a drug treatment of oxalic acid dripped on the family without any brood. Winter treatment is carried out with amitraz-based drugs by 17 (40.5%) of respondents, with the application of oxalic acid after queen caging, as declared by 13 (30.9%) of the beekeepers; the remaining 10 (23.8%) state that they carried out repeated cycles of sublimated oxalic acid until they no longer detect *Varroa* fall on the diagnostic bottom board; 2 beekeepers (4.8%) apply oxalic acid without caging the queen, trying to intercept a natural oviposition interruption. Veterinary drug rotation was carried out by 29 (69%) of the interviewed beekeepers.

### 3.3. Statistical Analysis (Management Practices)

The management practices implemented that have been correlated with the recorded level of infestation are summarized in Table 7.

Analysis of variance (ANOVA) showed that certain management practices are more influential on the infestation rate of apiaries. Among the management practices considered (Table 6), queen replacement and drone brood removal are two management techniques that, when used, are linked to lower levels of infestation. This result is statistically significant (*p* < 0.05) when comparing the infestation rates of apiaries that carry out these operations with those that do not. A lower infestation level is also associated with a higher number of drug treatments administered (*p* < 0.05). The other GBPs considered, reported in Table 6, did not have a statistically significant influence (*p* > 0.05) on the infestation level of the apiaries.

## 4. Discussion

Control of the *V. destructor* population is crucial because infestations increase honey-producing costs [36]. It is important to arrive in spring with a low infestation level to avoid significant losses in autumn. *V. destructor* populations can be controlled with a variety of techniques, chemicals, or natural compounds. The most time-efficient treatments involve the use of products based on chemical active ingredients [37]. However, *V. destructor* are becoming resistant to fluvalinate, coumaphos, and amitraz [38,39,40]. The low sensitivity to miticides often leads to the overuse of drugs, which can result in an accumulation of these substances or their metabolites in honeybee colonies [41]. The adoption of IPM techniques for *Varroa* mite control could be a successful approach [42]. Indeed, IPM techniques comprise a series of actions involving proactive behavior throughout the year aimed at prevention rather than cure [17]. However, each operation’s efficiency varies and is unpredictable. Our study aimed to shed light on the influence that the actions implemented can have on the degree of infestation. The data obtained from our investigation were correlated with the altitude values of the areas.

The results of this study indicate that the application of good beekeeping practices helps to keep the level of infestation below the damage threshold. Management practices, such as queen replacement [43] and drone brood removal [44], have a positive impact, leading to a reduction in the infestation rate. These actions were most often implemented by professional beekeepers. The same behaviors do not occur in non-professional beekeepers. The low use of these IPM practices on these apiaries is often due to a lack of knowledge of these techniques. Seventy percent of the beekeepers surveyed reported they do not remove drone broods because they do not know about the procedure or think it is ineffective, for instance. Some of these beekeepers also declare that they do not systematically diagnose *V. destructor* infestation on samples of adult honeybees. These beekeepers use the simple observation of the natural fall on the diagnostic bottom board, a method suggested for research/selection purposes, and that is unsuitable for diagnosis if not carried out systematically and with care [19].

Another finding of the study is the influence of the number of treatments on the percentage of infestation. The latter appears to be obvious and predictable evidence but highlighting it may help to reconsider intervention strategies in certain geographical areas. Since 2016, the Ministry of Health in Italy has annually published “Guidelines for the control of *V. destructor* infestations” developed by the CNR (Centro di Referenza Nazionale—Istituto Zooprofilattico delle Venezie) [45]. The purpose of these guidelines is to offer advice on the usage of pharmaceutical drugs. According to the Ministry of Health, pharmacological treatments are one of the key control methods, as long as they are carried out taking into account the *V. destructor* reproductive cycles. Moreover, treatments must be administered in a capillary manner throughout the region according to pre-established timetables [46]. Due to the well-known phenomenon of re-infestation [47], which makes the control of this pest more challenging, the efforts of individual beekeepers must be linked and coordinated. The aforementioned “Guidelines for the control of *V. destructor* infestation” state that depending on the climatic conditions, treatment must be administered “at least” twice a year. To this, “at least” should be given special attention. This information indicates that other treatments may be necessary in addition to the two listed, depending on different geographical and meteorological variables that are also related to the severity of the infestation and nectar flows. Further interventions may be necessary also in case of incorrect use of veterinary drugs and/or re-infestation phenomena due to uncoordinated treatments between beekeepers. Chemurot et al. (2016) showed that mite infestation levels decreased with increasing altitude [22]. The lack of difference in infestation levels between the geographical areas (A1, A2, and A3) examined in the study could depend on the number of treatments carried out by beekeepers. There is also another element to take into account. Years ago, a natural autumn–winter laying gap occurred in the national area. At the geographical coordinates in which the Calabria region falls, winters are also quite mild. Partly as a result of the increase in global temperature [48], the natural blockage of oviposition that occurred in past years has disappeared. Therefore, it can be said that *Varroa* populations grow similarly and without interruption between the different altitude layers in Mediterranean-type climates.

The infection level of the colony doubles approximately every month when the brood is present. The mite increases significantly and reaches critical levels within a few months as a result of this exponential dynamic. To reduce the burden of colony infestation and allow adequate development of overwintering bees, at least one new intervention could be required to control the expansion of the *V. destructor* population [49]. For all these reasons, more than two treatments might be indicated in this geographical area. However, treatments must be started following a diagnosis. In this way, overuse of the drug would be avoided. Post-treatment diagnosis also allows for the verification of any emerging resistance phenomena. Information gathered in this way can help health authorities recommend the best treatment solutions.

A further consideration concerns the rotation of drugs. Beekeepers that claimed to practice drug rotation were essentially not introducing new active ingredients. In fact, the rotation consisted of replacing an organic acid (oxalic acid) with another (formic acid) in the summer period. Only 4.7% of beekeepers employ essential oils, preventing them from enhancing the arsenal of useful chemicals for treating ailments.

In Italy, the involvement of veterinarians in beekeeping is lower than in other animal husbandry. For this reason, hives are managed somewhat “autonomously”, both in terms of management and the selection of veterinary drugs for the treatment of *V. destructor*. Consequently, it can be argued that more training for beekeepers is needed. The support and presence of specialized figures in these realities are more than necessary [50]. In addition, the introduction of new acaricides characterized by high efficacy and low environmental impact is imperative [51,52].

Considering this evidence, veterinarians or beekeeping technicians should recommend the application of GBPs. For example, the treatments should be administered in combination with bio-techniques, such as queen caging, and the use of drugs that are unlikely to lead to the development of resistance, such as those based on essential oils, should be preferred [18,53,54,55,56].

## 5. Conclusions

Our study leads us to establish that GBPs are essential, and the implementation of the role of veterinarians is mandatory for the sector. Treatments are often administered without assessing the level of infestation. Furthermore, the efficacy of acaricidal treatment is not evaluated as is the possibility of re-infestation phenomena. Additionally, in Italy, where the veterinary drug in beekeeping is supplied without a prescription, the misuse and mismanagement of drugs make it urgent that the sale of pharmaceutical treatments follow a medical prescription. Only in this way can the beekeepers receive indications to remedy behavior that is unsuitable for successful parasite control.

## Figures and Tables

**Figure 1 microorganisms-11-01228-f001:**
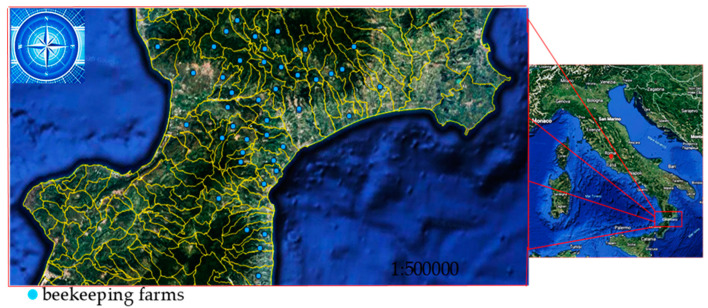
Locations of the beekeeping farms in the Calabrian region of Southern Italy, in which honeybee samples were taken to determine V. destructor load.

**Figure 2 microorganisms-11-01228-f002:**
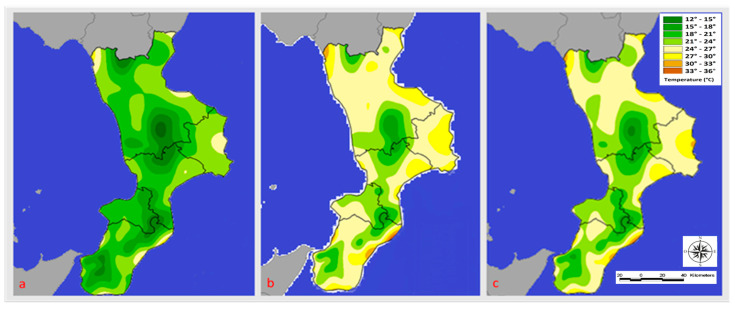
Temperature (°C) registered in June (**a**), July (**b**), and August (**c**) 2020.

**Figure 3 microorganisms-11-01228-f003:**
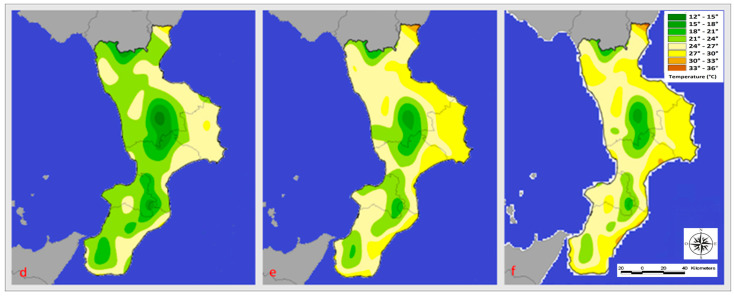
Temperature (°C) registered in June (**d**), July (**e**), and August (**f**) 2021.

**Table 1 microorganisms-11-01228-t001:** Subdivision of the municipalities into three groups, in relation to the altitude bands above sea level: A1: 150–350 m; A2: 350–750 m; A3: 700–900 m.

Groups	Municipalities
A1 (100–350 MASL)	Borgia, Catanzaro, Feroleto Antico, Isca sullo Ionio, Guaradavalle, Lamezia Terme, Maida, Marcellinara, Pianopoli, San Floro, Sellia marina, Squillace, Vallefiorita, Badolato
A2 (350–700 MASL)	Amaroni, Caraffa, Chiaravalle, Cortale, Curinga, Girifalco, Gimigliano, Falerna, San Sostene, Sellia, Stalettì, Jacurso, Zagarise, Belcastro
A3 (700–900 MASL)	Carlopoli, Decollatura, Petronà, Sersale, Tiriolo, Serrastretta, San Pietro Apostolo, Pentone, Platania, Soveria Mannelli, Cerva, Cicala, Fossato Serralta, Andali

**Table 2 microorganisms-11-01228-t002:** Results of the *V. destructor* diagnosis in A1 group (150–350 meters), A2 group (350–700 m) and A3 group (700–900 m) with the number of positive apiaries (above the threshold of 3%), prevalence (%) mean percentage of infestation registered in summer (a) and winter (b) diagnosis (years 2020 and 2021), standard deviation (SD) and standard error (SE).

a								
**Groups**	Year 2020				Year 2021			
	Positive Apiaries	Prevalence (%)	Mean Infestation (%)	SD (±)	Positive Apiaries	Prevalence (%)	Mean Infestation (%)	SD (±)
A1 (150–350 MASL)	7/14	50	3.12	1.75	8/14	57.1%	3.49%	1.76
A2 (350–700 MASL)	8/14	57.1	3.87	2.79	8/14	57.1%	3.36%	1.53
A3 (700–900 MASL)	8/14	57.1	4.31	2.59	5/14	35.7%	3.17%	2.70
**b**								
**Groups**	**Year 2020**				**Year 2021**			
	**Positive Apiaries**	**Prevalence (%)**	**Mean Infestation** **(%)**	**SD (** **±)**	**Positive Apiaries**	**Prevalence (%)**	**Mean Infestation** **(%)**	**SD (** **±)**
A1 (150–350 MASL)	13/14	92.8%	3.86%	0.87	9/14	64.2%	3.10%	1.13
A2 (350–700 MASL)	8/14	57.1%	3.02%	0.72	9/14	64.2%	3.55%	1.82
A3 (700–900 MASL)	4/14	28.5%	2.56%	1.31	6/14	42.8%	3.12%	1.63

**Table 3 microorganisms-11-01228-t003:** Mean infestation percentage in the tested apiaries before and after summer and winter treatment in 2020 and 2021.

Mean (%)	Group	N	Mean Infestation (%)	SD (±)
Before summer treatment After summer treatment	2020 2020	42 42	3.774 * 0.646 *	2.416 0.379
Before summer treatment After summer treatment	2021 2021	42 42	3.343 * 0.621 *	2.020 0.412
Before winter treatment After winter treatment	2020 2020	42 42	3.151 * 0.693 *	1.122 0.510
Before winter treatment After winter treatment	2021 2021	42 42	3.262 * 0.503 *	1.536 0.366

* *p* value < 0.01.

**Table 4 microorganisms-11-01228-t004:** Schematic representation of the survey results (general characteristics).

Varroa-Control Factor (n. Apiaries = 42)	N	%
Beekeeper’s experience		
Practicing beekeeping for more than 5 years	22	52.4
Practicing beekeeping for less than 5 years	20	47.6
Number of hives per apiary		
More than 50	27	64.2
Between 30 and 50	10	22.7
Less than 30	5	11.3
Alternation of hive colors		
Yes	21	50
No	21	50

**Table 5 microorganisms-11-01228-t005:** Schematic representation of the survey results (management practices).

Varroa-Control Factor (n. Apiaries = 42)	N	%
Drone brood removal		
Yes	13	29.5
No	31	70.5
Frequency of queen replacement		
Every year	5	1.9
Every two years	8	19
Every three years	0	7.5
No replacement	29	69
Techniques employed for diagnosis		
Natural mite fall observed on the bottom board	15	35.7
Alcohol wash	7	16.6
Sugar roll test	14	33.3
Carbon dioxide	0	0
Brood inspection	6	14.2

**Table 6 microorganisms-11-01228-t006:** Schematic representation of the survey results (treatments).

Varroa-Control Factor (n. Apiaries = 42)	N	%
Number of treatments per year		
One	0	0
Two	15	35.8
≥Three	27	64.2
Active ingredients of pharmaceutical preparations and associated techniques used for summer treatment		
Amitraz	13	30.9
Formic acid	5	11.9
Oxalic acid applied after queen caging	15	35.7
Oxalic acid applied without caging and amitraz	6	14.2
Mix of oxalic acid and monoterpenes of essential oils	0	0
Mix of oxalic acid e formic acid	0	0
Monoterpenes of essential oils	2	4.7
Flumethrin	0	0
Oxalic acid sublimation	1	2.3
Active ingredients of pharmaceutical preparations and associated techniques used for winter treatment		
Amitraz	17	40.5
Oxalic acid sublimation	10	23.8
Oxalic acid applied after queen caging	13	30.9
Oxalic acid applied without caging	2	4.8

**Table 7 microorganisms-11-01228-t007:** Management practices that have been assumed to influence parasite load the most.

Management Practice	Description
Alternation of hive colors	Reduction of drift phenomena
Queen replacement	Maintain young queen in hives
Drug	Periodic drug rotation
Monitoring program	Performing clinical diagnosis of infestation
Treatment	Number of acaricide treatments performed
Drone brood	Periodic elimination of the drones brood

## Data Availability

The data are kept at the University of Magna Græcia of Catanzaro and are available upon request.

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
