# Peer review of "Prevalence of Varroa destructor in Honeybee (Apis mellifera) Farms and Varroosis Control Practices in Southern Italy"

_microorganisms, 2023, doi:10.3390/microorganisms11051228_

Round 1

Reviewer 1 Report

The manuscript not clear, is relevant for the field but is presented in a not well-structured manner.The manuscript is scientifically sound and is the experimental designis appropriated to test the hypothesis but the tables not properly show the data. The data not are interpreted appropriately and consistently throughout the manuscript, The conclusion are consistent with the evidence and arguments presented.

Author Response

Reviewer 1

The manuscript not clear, is relevant for the field but is presented in a not well-structured manner. The manuscript is scientifically sound and is the experimental designs appropriated to test the hypothesis but the tables not properly show the data. The data not are interpreted appropriately and consistently throughout the manuscript. The conclusions are consistent with the evidence and arguments presented.

R: we thank the reviewer for the thorough examination of the article and the suggestions that help us improve its overall quality. In accordance with suggestions, we have modified the tables and presented the data more clearly. In particular, the answers to the questionnaire have been grouped by type and thus 3 macro-sections are now created: 1) general characteristics of the apiary, 2) management practices, 3) treatments. Moreover, a paragraph entitled “Statistical analysis (management practices)” has been created in which the evidence from the statistical processing of the collected data was included.

Reviewer 2 Report

Dear authors,

You submited a manuscript entitled '' Prevalence of Varroa destructor in honeybee (Apis mellifera) farms and varroatosis control practices in Southern Italy.'' 

In your article I can not find any new information about the varroa infection, but I can find many serious mistakes about the design and the implementation of your experiment. Below you will find my corrections about your manuscript. The idea of your experiment was very good but you focus on wrong points, which give no new information. In many cases your sentences make no sence. 

Line 55: What do you mean with ''High regard''? 

line 66-67: Varroa destructor you must add Anderson and Trumman 

line 96: I think it is 3% and not 2% and it is for brood absence periods (not only for drone absence) 

linbe 137: Probably fourth and not quarter

lines 144-170: these all are results and not materials and methods

lines 176-177: Why did you select small apiaries. Please explain

Line 179: enrolled of envolved ?

Line 211: I do not think that sugar roll test is the best way to determine the varroa infection (please give more information in materials and methods) 

Line 233: beekeepers and not farmers (correct it in manuscript)  

line 261: What do you mean with the positive sample? please determine the positive sample in materials and methods section 

line 274: Positive apiaries not farms (correct it in manuscript) 

Table 5: Bottom board observation. what is it? please explaine it in materials and methods with the other methods that are described in the table. In my opinion observing the bottoms for sure is not the correct way to say that you have a positive sample. There is a natural fall of varroa (with higher or lower rate) in all hives. Also, there is natural infection in all hives, so this method is unsafe to check the efficacy of a treatment. 

line 350: give reference

line 371-373: Nothing new. Applying a treatment (beekeeping practice) you will keep the infection level low. 

Lines 382-383: Nothing new. The application with acaricides many times nedds more repetitions, but there are acaricides with high efficacy with only one repetition. So you can not say something about the number of repetitions and the infection level. 

lines 410-411: Nothing new. It is known that the presence of brood make the infection level higher, because the biological cycle of varroa takes place in bee - brood

Lines 428-429: Not only the training of beekeepers is necessary, but also the new acaricides with high efficacy must be introduced (which is the biggest problem in beekeeping).

Line 431: Consindering these considerations? 

Author Response

Reviewer 2

Dear authors,

You submited a manuscript entitled '' Prevalence of Varroa destructor in honeybee (Apis mellifera) farms and varroatosis control practices in Southern Italy.''

In your article I can not find any new information about the varroa infection, but I can find many serious mistakes about the design and the implementation of your experiment. Below you will find my corrections about your manuscript. The idea of your experiment was very good but you focus on wrong points, which give no new information. In many cases your sentences make no sence.

R: We thank the reviewer for his comments and suggestions that helped to improve the manuscript as a whole. Although no evidence of particular relevance to the field has emerged, we believe that the manuscript is nevertheless of important value. Indeed, this is the first time that an epidemiological study has been conducted on Varroa destructor in the Calabria region of southern Italy, while analysing the management practices implemented by Calabrian beekeepers to control the parasitosis. It also emphasises, with supporting scientific evidence, the importance of applying good beekeeping practices.

The English has been extensively revised and corrected by a native speaker, so that the text would be perfectly understandable to readers. We also responded to all the suggestions made in detail, as can be seen below

Line 55: What do you mean with ''High regard''?

R.: The sentence has been changed in accordance with the suggestion

line 66-67: Varroa destructor you must add Anderson and Trumman

R.: Changed as suggested

line 96: I think it is 3% and not 2% and it is for brood absence periods (not only for drone absence)

R.: We thank the reviewer for this observation, which allowed us to add a bibliographical note and allowed us to notice a typo, as the word “removal'” was missing next to “drone brood”. The 2% is correct as we referred to the percentage of infestation in spring, when the drone brood removal is mainly applied. The added bibliographical note refers to the guide published by Vetòpharma where you can consult the table with the infestation percentages not to be exceeded in relation to the periods of the year.

linbe 137: Probably fourth and not quarter

R.: many thanks, it has now been modified as suggested

lines 144-170: these all are results and not materials and methods

R.: We have transferred this section to the results, as suggested

lines 176-177: Why did you select small apiaries. Please explain

R.: We have provided an explanation in the text and added a bibliographical note

Line 179: enrolled of envolved ?

R.: modified as suggested

Line 211: I do not think that sugar roll test is the best way to determine the varroa infection (please give more information in materials and methods)

R.: We have provided explanations in the text and added bibliographical notes to support the decision

Line 233: beekeepers and not farmers (correct it in manuscript) 

R.: modified throughout the manuscript as suggested

line 261: What do you mean with the positive sample? please determine the positive sample in materials and methods section

R.: We have added this information in the materials and methods

line 274: Positive apiaries not farms (correct it in manuscript)

R.: modified throughout the manuscript as suggested

Table 5: Bottom board observation. what is it? please explaine it in materials and methods with the other methods that are described in the table. In my opinion observing the bottoms for sure is not the correct way to say that you have a positive sample. There is a natural fall of varroa (with higher or lower rate) in all hives. Also, there is natural infection in all hives, so this method is unsafe to check the efficacy of a treatment.

R.: We have explained further in the table that it is the observation of the natural fall on the diagnostic bottom board. We too are of the opinion that observation of the diagnostic bottom board is not a good method of checking for parasite load. In the manuscript, it was only pointed out that some beekeepers practice it, as stated in the questionnaire administered. We appreciated your suggestion that led to this revision with the addition of a sentence and a bibliographical note in the “considerations” clarifying that this method implemented by some beekeepers is not appropriate.

line 350: give reference

R.: This is a result that emerged from our statistical analysis of the data. We have amended the manuscript and added a paragraph to make the following statement more understandable

line 371-373: Nothing new. Applying a treatment (beekeeping practice) you will keep the infection level low.

R.: We thank the reviewer for this comment, which allowed us to clarify a concept that we consider important in the manuscript. What we would like to emphasise is that some beekeepers practise more than two treatments at these latitudes and longitudes and, therefore, in the climatic conditions typical of the Mediterranean area. As stated in the conclusions and by the Italian Ministry of Health, the climatic variations from North to South in the country are considerable, so more than two treatments may be necessary in some areas of the country. This is what we want to bring out and which we have now clarified in the conclusions.

Lines 382-383: Nothing new. The application with acaricides many times needs more repetitions, but there are acaricides with high efficacy with only one repetition. So you can not say something about the number of repetitions and the infection level.

R.: We thank the reviewer for this consideration, which allowed us to clarify the concept in the manuscript. Some beekeepers practice treatments inappropriately and do not associate the treatment with bio-techniques. This is why they resort to multiple treatments. This has now been better clarified in the text.

lines 410-411: Nothing new. It is known that the presence of brood make the infection level higher, because the biological cycle of varroa takes place in bee - brood

R.: The sentence has been rephrased to make it easier to understand. The concept is a premise for the the clarification of a possible need for more than two treatments in this area

Lines 428-429: Not only the training of beekeepers is necessary, but also the new acaricides with high efficacy must be introduced (which is the biggest problem in beekeeping).

R.: We find this comment particularly valuable. This consideration was added to “Discussion” section of the manuscript.

Line 431: Consindering these considerations?

R.: the sentence has been rewritten to make it easier to understand

Reviewer 3 Report

Major revision: 1. The authors stated that “The questionnaire consisted 235 of 30 questions” but they did not demonstrate what the exactly are 30 questionnaires? 2. Authors have to clearly state how to control effectively varroa infection at a low level by current investigation instead of  through integrated pest management programs.  

Minor revision

1. The resolution is not enough, especially figure notes can not be seen such as figure 1.

Author Response

Reviewer 3

Major revision: 1. The authors stated that “The questionnaire consisted 235 of 30 questions” but they did not demonstrate what the exactly are 30 questionnaires? 2. Authors have to clearly state how to control effectively varroa infection at a low level by current investigation instead of through integrated pest management programs. 

R: we thank the reviewer for his suggestions. Only one survey of 30 questions was administered to beekeepers. We have clarified this concept in the manuscript.

It’s true that we emphasised the importance of integrated pest management programmes (IPM) in the text. However, must be said that all the practices analysed fall under these programmes. As suggested, we have now placed more emphasis on the results in the “considerations” section. Thus, the importance of keeping young queens in the apiary, making infestation diagnoses and practising drone brood removal has been emphasised.

Minor revision

The resolution is not enough, especially figure notes can not be seen such as figure 1.

R.: The figure has been modified as suggested.

Round 2

Reviewer 1 Report

For me so corrected manuscript  can be accepted.

I have nothing to add

Reviewer 3 Report

It can be accepted.